# Spatio-Temporal Pattern of Green Agricultural Science and Technology Progress: A Case Study in Yangtze River Delta of China

**DOI:** 10.3390/ijerph19148702

**Published:** 2022-07-17

**Authors:** Chen Qian, Caiyao Xu, Fanbin Kong

**Affiliations:** 1Institute of Ecological Civilization, Zhejiang A&F University, Hangzhou 311300, China; chenq@stu.zafu.edu.cn; 2Research Academy for Rural Revitalization of Zhejiang Province, Zhejiang A&F University, Hangzhou 311300, China; 3College of Economics and Management, Zhejiang A&F University, Hangzhou 311300, China

**Keywords:** green agricultural science and technology progress, driving mechanism, food security, carbon emission, Yangtze river delta

## Abstract

Green agricultural science and technology progress (GASTP) plays an important role in the green transformation of agriculture. This study calculates the contribution rate of GASTP by using the Super-SBM model in the Yangtze River Delta (YRD) from 2011 to 2020. The exploratory spatial data analysis (ESDA) method and the Fixed Effect (FE) panel data model method were adopted to empirically analyze the spatio-temporal patterns of GASTP and its driving mechanism in the YRD. The results showed that: (i) except for Shanghai from 2011 to 2015, the contribution rate of GASTP in the YRD was generally lower than 1 in Anhui Province, Jiangsu Province, and Zhejiang Province, (ii) the level of GASTP had a positive spatial correlation with the study period, except for 2017, and (iii) per capita GDP, agricultural mechanization level, agricultural financial support, and planting structure are four influencing factors of GASTP in the YRD, while total retail sales of social consumer goods and total exports did not have significant effects on GASTP in the YRD. Therefore, we need to increase the opportunities to exchange GASTP experience between cities, improve the environment for agricultural technology extension, and develop follow-up monitoring mechanisms.

## 1. Introduction

Food security and carbon emissions have attracted worldwide attention [1,2]. A report on “World Food Security and Nutritional Status in 2021” by the Food and Agriculture Organization of the United Nations (FAO) showed that about 811 million people worldwide faced food insecurity in 2020, accounting for 10.4% of the global population [3]. Constrained by the area of cultivated land, the demand for food production needs to rely more on technology progress to achieve better sustainability. The United Nations Intergovernmental Panel on Climate Change (IPCC) released “Climate Change 2022: Mitigation Climate Change”, noting that global carbon emissions reached 59 billion tons in 2019, which was a 12% increase in the global emissions of 52.5 billion tons in 2010 [4], whereas, in total emissions, the proportion of China’s agricultural CO_2_ is 17% [5]. Therefore, China has proposed a “double carbon” target in order to reduce agricultural CO_2_ emissions and improve agricultural technical efficiency.

Green agricultural science and technology progress (GASTP) refers to the contribution share of agricultural science and technology progress to the growth rate of agricultural output, which not only improves the production efficiency and reduces the production cost in agriculture but is also used as an important means to achieve high-quality development of agriculture [6,7]. Academic circles have mostly adopted the Cobb-Douglas (C-D) production function, the Solow residual method, and the data envelopment method (DEA). Wang et al. (2021) analyzed the Sichuan province of China by using the C-D production function [6]. The Solow residual method was measured in China [8] and the United States [9]. The DEA was used to calculate China’s contribution rate of agricultural technology progress from 1984 to 1993 [10]. These studies provide a helpful basis to understand and apply the methods used to measure the contribution rate of GASTP in this study.

For a long time, scholars have focused on technology progress. The famous classical economist Adam Smith first put forward the theory that technology progress can reduce labor and promote economic development [6]. Later, Schumpeter put forward the growth theory, which mainly discussed the relationship between technology innovation and economic growth. Its core feature was that endogenous technology progress and R&D innovation were the decisive factors to promote economic development [7]. Technology progress is the main driver of carbon reduction [11], and most of the existing research on technology progress has been concerned with carbon emissions. Zhong et al. (2022) found that it is possible to improve the digital economy and reduce carbon intensity by mediating technology progress in agriculture [12]. Ge (2022) took 13 divisions of the Xinjiang Production and Construction Corps as samples from 1998 to 2017 and showed that the impact of agricultural technology progress on agricultural carbon emission efficiency is affected by human capital and agricultural economic development level [13]. By evaluating the energy efficiency of maize cultivation techniques in different farming systems, Laura et al. (2014) found that corn cultivation technology can effectively reduce agricultural carbon emissions [14]. Furthermore, the researcher has analyzed China’s agricultural green technology progress from the perspective of time and space. Tian and Yin (2021) thought technology progress had a significant inhibitory effect on the carbon emission intensity of agricultural energy, due to a significant spatial spillover effect [15]. Deng and Cui (2022) used the panel data on 31 provinces in China from 1998 to 2018, adopted the EBM-GML model to measure the progress of China’s agricultural green technology, and explored the dynamic evolution characteristics of the spatio-temporal dimension of China’s agricultural green technology progress [7]. Throughout Chinese and overseas literature, it is clear that scientific and technology progress has been used to make up for deficiencies in agricultural infrastructure and the curbing of carbon emissions [16,17,18] as well as the lack of research in the analysis of the contribution rate of GASTP and its driving forces under carbon emission constraints. Hence, in the process of the global pursuit of green and high-quality agricultural development, the improvement of GASTP and the sustainable development of the agricultural environment will be of great significance [19].

Since the reform and opening up, China’s agricultural and economic development has been rapidly growing while also bringing high levels of resource consumption and pollution [20]. The excessive use of chemical fertilizer, agricultural film, and other production materials in the process of economic development leads to excessive agricultural carbon emissions and restricts the sustainable development of agriculture [21]. The extensive economic growth at the cost of resource consumption and environmental pollution is unsustainable. GASTP has become the mainstream method for changing production mode and improving production efficiency, and it is also the inevitable choice to achieve high-quality economic development [22]. Therefore, the central government has proposed to achieve green agricultural development. In October 2015, China put forward the five development concepts of “innovation, coordination, green, openness and sharing” and pointed out that “promoting the harmonious coexistence of man and nature and promoted the establishment of green low-carbon cycle development industry system”. The Outline of the 14th Five-Year Plan and the Long-Range Objectives Through the Year 2035 clearly stated that we should accelerate the green development of agriculture and continuously improve the rural ecological environment [23]. In 2021, the “Opinions on Promoting the Institutional Mechanism of Innovation for Green Agricultural Development”, issued by the General Office of the State Council, pointed out that the basic driving forces for green agricultural development should be technology innovation [24]. Considering that technology progress has an important impact on the rural ecological environment, it is important to study the spatio-temporal patterns of GASTP contribution rate and its driving mechanism in the Yangtze River Delta region under the constraint of carbon emissions. The YRD is located in the middle and lower reaches of the Yangtze River. As an agricultural resource, it is rich and diverse and occupies an important position in the overall pattern of agricultural development in China. Many studies have only focused on agricultural technology progress from the national level [7,12,25], and few studies have specifically analyzed agricultural technology progress at the city prefecture level, and the research on the contribution rate of GASTP in the YRD has been even more limited. Therefore, this study takes the YRD as the research object. Under the constraint of carbon emissions, the Super-SBM model was used to calculate the contribution rate of GASTP from 2011 to 2020, and then the ESDA model and the FE panel data model were combined to explore the spatio-temporal evolution characteristics and analyze its driving mechanism. This study combines carbon emission constraints with the contribution rate of GASTP in the YRD, which is beneficial to fill the knowledge gap in the YRD’s contribution rate of GASTP. Per capita GDP, agricultural mechanization level, agricultural financial support, and planting structure had a significant relationship with the contribution rate of GASTP, indicating that these factors promote GASTP in the YRD. Therefore, the evolution law of GASTP in the YRD was summarized, providing a reference for the agricultural green transformation in other regions of China.

The contributions of this research are as follows: compared with the traditional protection and governance of the agricultural environment, agricultural green development is a sustainable development model that integrates the environment into the economy and society. Further research can enrich GASTP theory. Moreover, as our analysis combines the carbon emission constraints and spatio-temporal patterns of GASTP, it can better reflect the reality of agricultural green technology progress in the YRD, and the analysis of the driving mechanism of GASTP is conducive to the formulation and implementation of policies.

## 2. Materials and Methods

### 2.1. Study Area

The YRD (28°45′–33°25′ N, 118°20′–123°25′ E) in this study includes Shanghai, Jiangsu, Zhejiang, and Anhui and consists of 41 cities (Figure 1), with a total area of about 358,000 km^2^, and the topography mainly consists of plains. By the end of 2020, the total population of the YRD exceeded 235 million, and GDP and agricultural added value reached 24.5 trillion yuan and 999.40 billion yuan, respectively, accounting for more than 12% of the total agricultural economy in China. In the “National Agricultural Sustainable Development Plan (2015–2030)”, three provinces and one city in the YRD were included in the optimal development area [26]. Therefore, agricultural development in the YRD will be crucial for China’s agricultural economy. However, in the process of agricultural production, due to the widespread use of agricultural mechanization, carbon emissions will be generated, so it is of particular concern to study the spatio-temporal patterns of the contribution rate of GASTP and its driving mechanism in the YRD under carbon emission constraints.

### 2.2. Methodology

#### 2.2.1. Measurement of the Contribution Rate of GASTP

Index Selection

In view of the availability of data, and combining the research results of the existing literature [27], seven evaluation indexes were selected from the input and output dimensions to construct the contribution rate of GASTP in the YRD. The specific characterization variables are shown in Table 1.

2.The Super-SBM Model Using Undesirable Output

In the process of agricultural economic development, undesirable outputs such as carbon dioxide and other gases are generated. Tone (2002) showed that the Super-SBM model using undesirable outputs has been widely used in the literature [29,30,31]. The Super-SBM Model using undesirable output calculated the contribution rate of GASTP in the YRD. The model can be expressed as follows.
(1)$$\min\rho =\frac{\frac{1}{\;m}\sum_{i=1}^{m}(\frac{\bar{x}}{x_{\mathrm{ik}}})}{\frac{1}{r_{1}+r_{2}}(\sum_{s=1}^{r_{1}}\frac{y^{d}}{y_{\mathrm{sk}}^{d}}+\sum_{q=1}^{r_{2}}\frac{\bar{y^{u}}}{y_{\mathrm{qk}}^{u}})}\;\left\{ \begin{array}{l} \bar{x}\geq \sum_{j=1,\neq k}^{n}x_{ij}\lambda_{j;}\bar{y^{d}}\leq \sum_{j=1,\neq k}^{n}y_{\mathrm{sj}}^{d}\lambda_{j};\\ \bar{y^{d}}\geq \sum_{j=1,\neq k}^{n}y_{\mathrm{qj}}^{d}\lambda_{j};\bar{x}\geq x_{k};\bar{y^{d}}\leq y_{k}^{d};\bar{y^{u}}\leq y_{k}^{u};\\ \lambda_{j}\geq 0;i=1,2,\ldots ,m;j=1,2,\ldots ,n;\\ s=1,2,\ldots ,r_{1};q=1,2,\ldots ,r_{2} \end{array}\right.$$
where n is the number of decision-making units (DMUs), m is the number of inputs, r_1_ and r_2_ represent the number of desired outputs and undesirable outputs, respectively, and x, y^d^, and y^u^ are the elements in the corresponding input matrix, desired input matrix, and undesirable output matrix. ρ is the value of GASTP; when ρ < 1, it means that the DMU is inefficient, and when ρ ≥ 1, it means that the DMU is effective. Max DEA software was used to calculate the contribution rate of GASTP in the YRD from 2011 to 2020.

#### 2.2.2. Spatio-Temporal Pattern of Contribution Rate of GASTP

Exploratory Spatial Data Analysis (ESDA) is an important research method in spatial econometrics. In this study, the ESDA method was used to reveal the overall spatial correlation and spatial agglomeration of the contribution rate of GASTP in the YRD [32].

Global Moran’s I index was used to measure the degree of spatial autocorrelation. Its expression is
(2)$${\mathrm{Global}\;\mathrm{Moran}\;}^{’}I=\frac{\sum_{i=1}^{n}\sum_{j=1}^{n}w_{\mathrm{ij}}(x_{i}-\bar{x})(x_{j}-\bar{x})}{s^{2}\sum_{i=1}^{n}\sum_{j=1}^{n}w_{\mathrm{ij}}}$$
where x_i_ and x_j_ represent city i and city j, respectively, $-\bar{x}$ is the arithmetic mean of the contribution rate of GASTP of all cities, n is the number of cities studied, and w_i,j_ is the neighboring weight matrix, which indicates the neighboring relationship between two cities. The global Moran’s I index reflects the degree of spatial dependence in the overall spatial scope, and the range of Moran’s I index is in [−1, 1]. If the value of I is greater than 0, this indicates a positive spatial correlation, while a value of 0 indicates that there is no spatial correlation, and a value of less than 0 indicates irrelevance, denoting a spatial negative correlation. In this study, Geoda software was used to perform a global spatial autocorrelation test based on the Queen second-order adjacency weight matrix.

Local Moran’s I index is used to measure and draw the LISA cluster, which measures the spatial correlation characteristics of the region and adjacent regions. The calculation formula is
(3)$${\mathrm{Local}\;\mathrm{Moran}}^{’}\;I=\frac{\left(x_{i}-\bar{x})\sum_{i\neq j}^{n}W_{\mathrm{ij}}(x_{j}-\bar{x}\right)}{s^{2}}$$

The LISA cluster is used to identify agglomeration areas of the contribution rate of GASTP in the YRD, and it is divided into four categories: High-High Cluster (H-H), Low-Low Cluster (L-L), High-Low Cluster (H-L), and Low-High Cluster (L-H).

#### 2.2.3. Panel Data Model

To better demonstrate the driving factors of the contribution rate of GASTP, a panel data model was used in this study [32]. The panel data model is established as shown in Equation (4).
(4)$$\mathrm{GASTP}=\alpha +\sum \beta_{i}z_{i}+\mu$$
where GASTP indicates the contribution rate of GASTP, α stands for constant term, $\beta_{i}$ represents the partial regression coefficient, $z_{i}$ represents the factors affecting GASTP, and μ is a random error term.

Referring to the literature on agricultural technical efficiency [33], six factors were selected to study their effects on the contribution rate of GASTP. In terms of socio-economic characteristics, these mainly included the level of economic development (measured by per capita GDP), agricultural financial support (measured by the proportion of local financial agricultural forestry and water affairs to local financial expenditures), total retail sales of social consumer goods, and total exports. The level of economic development is expressed by the logarithm of the per capita GDP to reflect the economic development of the YRD. From the perspective of agricultural production characteristics, these mainly referred to the influence of planting structure (measured by the sown area of grain crops) and agricultural mechanization level of GASTP (measured by the total power of agricultural machinery as a proportion of the number of employees in the first industry). Considering the presence of heteroscedasticity, the larger values in the variables have been logarithmically processed. The descriptive statistical analysis results of specific indicators are shown in Table 2.

### 2.3. Data

The study evaluated the contribution rate of GASTP and its driving mechanism in the YRD, using panel data on 41 cities from 2011 to 2020. Original data was derived from the “Jiangsu Statistical Yearbook”, “Zhejiang Statistical Yearbook”, “Anhui Statistical Yearbook”, “Shanghai Statistical Yearbook”, “China Animal Husbandry Yearbook”, “China Rural Statistics Yearbook”, and the “Seventy Years of Glory: A Compilation of Historical Statistics in Shanghai”. Some missing data were compensated by interpolation.

## 3. Results

### 3.1. Analysis of Spatio-Temporal Pattern

As can be seen from Figure 2, the contribution rate of GASTP in the YRD was generally low and exhibited a decreasing trend with fluctuations. The value of the contribution rate of GASTP in Anhui Province was lower than the average of the YRD, whereas that of Jiangsu was higher than or equal to the average. In Shanghai city, the contribution rate of GASTP was highest in 2011, and the value was higher than the average of the YRD before 2015. However, Shanghai’s contribution rate of GASTP was lowest in 2020, and the value was lower than the average of the YRD after 2015. Except for 2014, the value of the contribution rate of GASTP in Zhejiang Province was higher than or equal to the average of the YRD, and the value was lowest in 2014.

The results of the spatial feature are depicted in Figure 3. The contribution rate of GASTP in the YRD has an obvious spatial variation. In 2011, high contribution rate and medium contribution rate areas were mainly distributed in the Anhui Province and Jiangsu Province, while low contribution rate areas were mainly concentrated in the cities of Suqian, Taaizhou, Huainan, Tongling, Jinhua, and Wenzhou. In 2013, high contribution rate areas were similar to those in 2011. The cities of Lu’an and Quzhou shifted from medium contribution rate areas to low contribution rate areas. In 2015, low contribution rate areas were expanded to include the cities of Huaibei, Bozhou, Taizhou, and Nantong. Lu’an city shifted from low contribution rate areas to medium contribution rate areas. Yancheng and Yangzhou shifted from high contribution rate areas to medium contribution rate areas. In 2018, Chuzhou, Lu’an, Wuhu, Jiaxing, Xuancheng, and Shaoxing cities shifted from medium contribution rate areas to low contribution rate areas. Hefei city shifted from medium contribution rate areas to high contribution rate areas. Wuxi city shifted from high contribution rate areas to medium contribution rate areas. Shanghai city shifted from high contribution rate areas to low contribution rate areas. Taizhou city shifted from low contribution rate areas to medium contribution rate areas. In 2020, high contribution rate areas and medium contribution rate areas sharply decreased. Huaian, Nanjing, Suzhou, Huzhou, Hangzhou, Zhoushan, and Chuzhou cities shifted to high contribution rate areas, while Chizhou, Suuzhou, Shaoxing, and Ningbo city shifted to medium contribution rate areas. Low contribution rate areas included the rest of the areas of the YRD.

### 3.2. Analysis of Spatial Correlation Characteristics

Results are shown in Table 3. Except for 2017 and 2020, the Moran’s I index of the contribution rate of GASTP in the YRD was positive, affirmed through significance testing at the levels of 5% and 1%. Except for 2017, the Moran’s I index of the contribution rate of GASTP in the YRD in the remaining years was spatially positively correlated, indicating that there was a high agglomeration or a low agglomeration effect in the spatial distribution.

Figure 4 shows the LISA cluster diagram of GASTP in the YRD. The numbers of H-H were 1 and 2 in 2011 and 2013. The numbers of L-L were 4, 4, 1, and 2 in 2011, 2013, 2015, and 2018. H-H was mainly distributed in Wuxi and Changzhou in Jiangsu. L-L was mainly located in Liuan, Fuyang, Bozhou, and Huaibei, which are mainly mountainous cities in Anhui Province. The number of H-H disappeared after 2013, and the number of L-L reduced until nothing remained after 2015. The numbers of H-L were 1 in 2011, 2013, 2015, and 2020. The numbers of L-H were 1 in 2011 and 2013. H-L were mainly distributed in Suuzhou and Chuzhou of Anhui Province, and Suuzhou was close to Huaibei, indicating that GASTP in Suuzhou was at a high level. Chuzhou is located in the lower reaches of the Yangtze River plain and the hilly area of Jianghuai. It showed high GASTP, but it did not drive the high-efficiency levels of the surrounding cities. L-H was mainly distributed in Taaizhou, Jiangsu Province, which borders Wuxi and Changzhou. However, the number of L-H decreased after 2013.

### 3.3. Driving Mechanism Analysis

According to the F test and Hausman test, P values were less than 0.05, the original hypothesis was rejected, and the estimation results of the FE model, Random Effect (RE) model, and ordinary least squares (OLS) model were analyzed (Table 4). From the regression results of the FE model, it can be seen that the four aspects of per capita GDP, agricultural mechanization level, agricultural financial support, and planting structure had a significant impact on the contribution rate of GASTP. Total retail sales and total exports of social consumer goods to the contribution rate of GASTP were not significant. Among them, only the planting structure had a positive impact on the contribution rate of GASTP, while per capita GDP, agricultural financial support, and agricultural mechanization level had a negative impact on the contribution rate of GASTP.

First, per capita GDP was negatively correlated with the contribution rate of GASTP, at a significance level of 5%, and showed that there was an “inverted U-shaped” relationship between per capita GDP and the contribution rate of GASTP. Therefore, the contribution rate of GASTP was represented by a downward trend with economic development.

Second, there was a negative correlation between agricultural mechanization level and the contribution of GASTP in the YRD, at a significance level of 10%. In the process of agricultural production activities, agricultural machinery will produce carbon emissions to a certain extent, polluting the environment.

Third, agricultural financial support had a significant negative correlation of 10% to the contribution rate of GASTP in the YRD, which was inconsistent with the expected results. The possible reason for this is that the structure and efficiency of government financial support for agriculture was not reasonable.

Fourth, planting structure was positive and significant at a level of 10%. The YRD had a superior regional position, a developed level of economic development, and important grain production based in Taihu Plain and Jianghuai Basin, so planting structure can have a positive impact on GASTP.

Finally, the measurement results of total retail sales and total exports of social consumer goods on the contribution rate of GASTP showed no obvious impact. In the YRD, the secondary industry and tertiary industry developed rapidly, so the total retail sales and total exports of social consumer goods on the contribution rate of GASTP were not obvious.

## 4. Discussion

### 4.1. Spatio-Temporal Pattern under Carbon Emission Constraints

Shanghai’s contribution rate of GASTP changed from a high contribution rate to a low contribution rate from 2011 to 2020. However, this outcome was contrary to that of Deng and Cui et al. (2022) who found that Shanghai was in a state of agricultural green technology progress only in 2011, 2014, and 2018 [7]. This difference could be attributed to the lack of consideration for carbon emissions. Interestingly, the contribution rates of GASTP for Huaian, Nanjing, and Suzhou cities were observed in a state of GASTP from 2011 to 2020, and those for Zhoushan, Ningbo, Huangshan, and Suuzhou were also in a state of GASTP except for 2017 and 2020. The plains may have played an important role in the contribution rate of GASTP because they are suitable for the developmentof green agriculture, such as in the areas around the cities ofHuaian, Nanjing, and Suzhou. While Zhoushan had been actively promoting the development of digital three rural areas and creating a smart pastoral, Ningbo had been shown great dynamism with its continuous rapid economic development since the reform and opening up, and it had successfully become a city of agricultural quality and safety in 2018. The fact that these places are at the center of economic development, coupled with policy and economic support, increased the efficiency of agricultural production and helped cultivate agriculture.

It is important for contemporary research to emphasize low carbon and to use efficient agricultural production technology to realize agricultural modernization. Therefore, those areas with low levels of development should integrate local agricultural resources, strengthen exchanges with neighboring areas, and draw on advanced production technology and management experience. Furthermore, it is important to pay more attention to the ecological environment and promote the coordinated development of the agricultural economy [34].

### 4.2. Spatial Correlation Characteristics

First of all, Wuxi and Changzhou are located in the center of the YRD and are in the hinterland of the plain. That Wuxi and Changzhou showed a positive spatial correlation is likely to be related to the fact that they have a good foundation for development and can radiate to surrounding cities. Second, Lu’an belongs to the northern foot of the Dabie Mountains. Fuyang and Bozhou can be affected by the floods under the Yellow River. There is no obvious location advantage that may affect the contribution rate of GASTP. Third, Suuzhou is close to Huaibei, indicating that Suuzhou is at a high GASTP. Chuzhou is located in the plains of the lower reaches of the Yangtze River and the hilly area of the Jianghuai region. It has obvious geographical advantages but has not driven the high GASTP level of surrounding cities. Finally, Taaizhou borders the cities of Wuxi and Changzhou where H-H is concentrated. Both Wuxi and Changzhou are in a state of high GASTP and do not radiate to Taaizhou, indicating that cities with high GASTP need to be further improved by driving surrounding cities with low GASTP. This requires that economically developed regions introduce international and Chinese advanced agricultural production technologies while developing the secondary and tertiary industries so as to ensure food production, increase farmers’ income, and create a beautiful environment. These effects may radiate to the surrounding areas, driving the agricultural resources of economically underdeveloped areas to the development direction of high output, low input, low emission. Moreover, this study revealed that the disparity in the contribution rate of GASTP between one city and its neighbors is increasing because of the disappearance of some clusters. The disappearance of H-H is a good illustration of the low-value observations that appeared [31]. 

### 4.3. Driving Mechanism Analysis

This study set out with the goal of exploring the evolution trend and optimization path of the contribution rate of GASTP in the YRD. In the analysis of the influencing factors on the contribution rate of GASTP and its trends, we found that per capita GDP, agricultural mechanization level, and agricultural financial support all correlated negatively with the contribution rate of GASTP, which are partly consistent with the findings of other studies. Therefore, there is a definite need for concern about the environmental problems caused by agricultural production because when the level of economic development increases, the demand for high-quality agricultural products and their living environment rapidly increases.

The government should take measures to promote the development and use of cleaner and environmentally friendly agricultural production techniques based on low carbon. Coupled with the increased education and environmental awareness of farmers, the use of environmentally friendly production technologies and methods of production can promote a shift from extensive agricultural production to intensive management. It is necessary that to improve the contribution rate of GASTP the government needs to adjust the quality and efficiency of fiscal expenditure [35] by developing targeted agro-environmental management policies. Some changes need to be made for the government to increase financial support for agriculture and raise the level of agricultural infrastructure construction, such as farmland water conservancy to improve agricultural production conditions. Moreover, it is beneficial that the effective implementation of agricultural environmental management activities be maintained to ensure agricultural pollution control and improve the efficiency of agricultural environmental governance. 

Even if agricultural machinery produces carbon emissions to a certain extent in the process of agricultural production activities, polluting the environment [31], strengthening the innovation, research, and development of agricultural mechanization products is meaningful. Furthermore, the local government is urged to innovate and research agricultural mechanization products that conform to the characteristics of China’s agricultural production. This could not only promote the development of agricultural machinery toward more intelligent and green technologies but also strengthen the integration of agricultural machinery and technology with agronomy, mobilize the enthusiasm of agricultural producers in using agricultural machinery, and promote the application and popularization of agricultural machinery. The more intelligent and green the agricultural machinery, the greater the enthusiasm agricultural producers will have in using agricultural machinery.

### 4.4. Implication and Limitation

GASTP is an important means to achieve high-quality agricultural development. According to the results of this study, policymakers can adopt strategies according to local conditions, which play an important role in further promoting the agricultural green policy system in the YRD. First of all, the government should establish a regular multilateral exchange system to strengthen the technical exchanges between the YRD and achieve the effect of shortening the GASTP level between cities. Then, the cities with low GASTP level need to improve their technology promotion system to create a good environment for promoting technology, in order to avoid the growing gap with high GASTP level cities. Finally, the government needs to establish a follow-up monitoring mechanism to supervise that the newly developed agricultural machines do not emit too much pollution, and do not ignore the agricultural environment because of blindly pursuing a high GASTP level. Only by coexisting in harmony with nature can we humans survive for a long time.

The disadvantage of this study is the data which comes from the statistical yearbook. The missing data has been calculated by imputation, which may add ambiguity and uncertainty to the analysis. Therefore, more detailed data should be collected in the following research, which can accurately reflect the status of green agricultural development, and allow policymakers to formulate an agricultural policy system that is more in line with the actual situation in the YRD.

## 5. Conclusions

This study added the carbon emissions occurring in the agricultural production process to the research framework of the contribution rate of GASTP, and used the Super-SBM model with undesirable outputs to calculate the contribution rate of GASTP in the YRD from 2011 to 2020, to explore the spatial and temporal evolution patterns of the contribution rate of GASTP in the YRD during this period. Finally, the FE model was applied to test the driving factors of the contribution rate of GASTP in the YRD. The results of the study indicated that:

First, the contribution rate of GASTP in the YRD was generally lower than 1, indicating that most of the prefecture-level cities in the YRD had a non-effective contribution rate of GASTP, and the overall level of these prefecture-level cities still had some room for improvement. In terms of spatial evolution patterns, the high contribution level of GASTP in the YRD is mainly distributed in northeast Zhejiang, south and middle Jiangsu and Huangshan, and Suuzhou in Anhui, while low efficiency exists in the southwest of Zhejiang.

Second, in terms of spatial correlation, between 2011 and 2020, except for 2017, there was a positive spatial correlation in the level of the contribution rate of GASTP in the YRD. This indicates that there was a clustering effect (H-H or L-L) in the spatial distribution.

Third, in terms of driving factors, per capita GDP, agricultural mechanization level, and agricultural financial support had a negative effect on the contribution rate of GASTP, planting structure had a positive effect on the contribution rate of GASTP, and total retail sales of social consumer goods and total exports had no obvious impact on the contribution rate of GASTP in the YRD.

## Figures and Tables

**Figure 1 ijerph-19-08702-f001:**
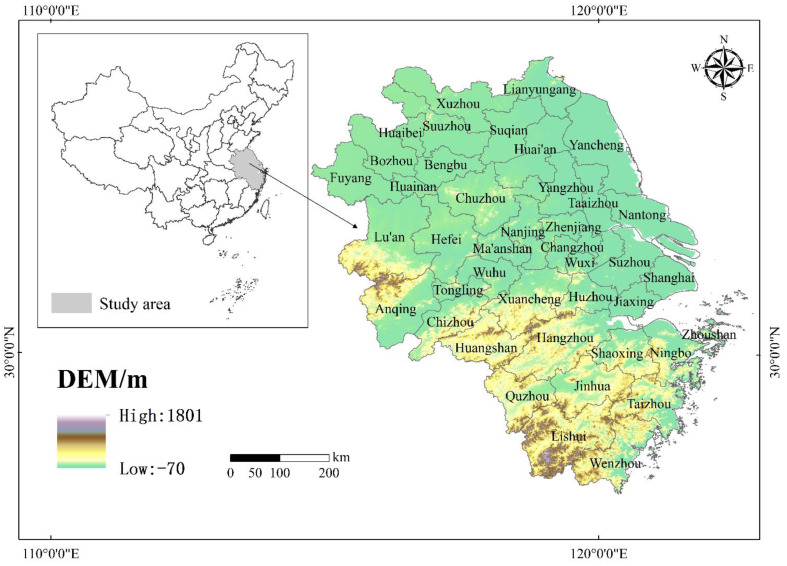
The location of the study area.

**Figure 2 ijerph-19-08702-f002:**
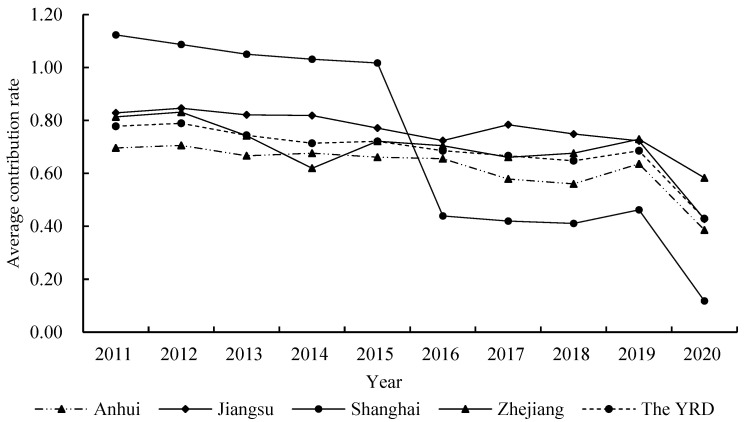
Average contribution rate of GASTP in the YRD and its provinces from 2011 to 2020.

**Figure 3 ijerph-19-08702-f003:**
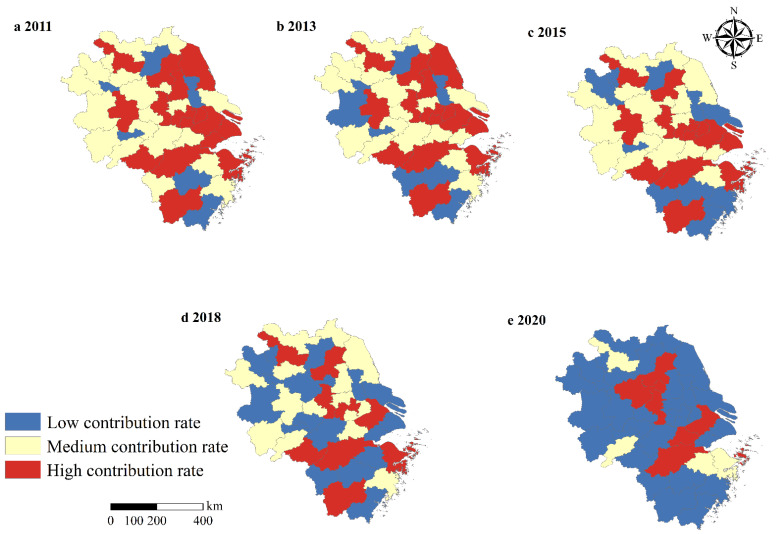
Spatio-temporal patterns of contribution rate of GASTP in 2011, 2013, 2015, 2018, and 2020.

**Figure 4 ijerph-19-08702-f004:**
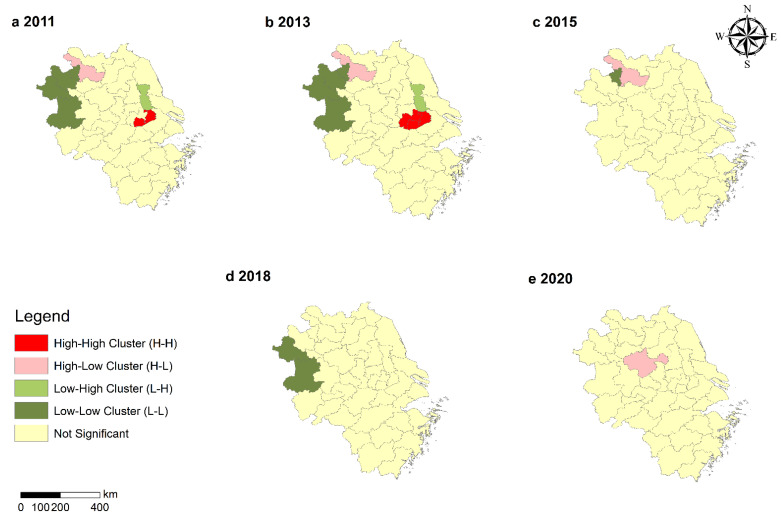
LISA of the contribution of GASTP in 2011, 2013, 2015, 2018, and 2020.

**Table 1 ijerph-19-08702-t001:** Evaluation indexes of the contribution rate of GASTP.

Index Types	First Level Index	Variable Declaration
Inputs	Labor	Number of employees in the first industry (million people)
	Capital	Total power of agricultural machinery (millions of kw)
	Land	Total sown area of crops (hectare)
	Energy	Rural electricity consumption (millions of kw/h)
	Water	Effective irrigation area (hectare)
Desirable output	Economic	Total output value of agriculture, forestry, animal husbandry, and fishery (billion yuan)
Undesirable outputs	Exhaust	Sum of carbon emissions from agricultural plastic film use, diesel use, pesticide use, year-end cattle stock, year-end pig stock, and year-end sheep stock. The calculation method is calculated by drawing on the practices of Tian Yun [28] and others (t)

**Table 2 ijerph-19-08702-t002:** Descriptive statistics of driving factors indicators.

Variables	Mean	Std. Dev	Minimum	Maximum
Per capita GDP	15.90	0.86	13.45	17.46
Agricultural financial support	0.11	0.05	0.03	0.86
Total retail sales of social consumer goods	10.01	1.41	6.74	13.96
Total exports	211.22	421.83	0.91	2102.77
Planting structure	5.54	3.04	1.35	25.87
Agricultural mechanization level	0.67	0.45	0.1	9.13

**Table 3 ijerph-19-08702-t003:** Moran’s I index of contribution rate of GASTP from 2011 to 2020.

Year	I	*p*-Value	Z-Value	E [I]	Mean	Sd
2011	0.136	0.008	2.785 ***	−0.025	−0.023	0.057
2012	0.155	0.004	3.143 ***	−0.025	−0.024	0.057
2013	0.141	0.007	2.892 ***	−0.025	−0.024	0.057
2014	0.134	0.012	2.812 ***	−0.025	−0.025	0.056
2015	0.135	0.006	2.738 ***	−0.025	−0.022	0.058
2016	0.110	0.019	2.350 **	−0.025	−0.023	0.057
2017	−0.076	0.181	−0.910	−0.025	−0.025	0.056
2018	0.141	0.007	3.011 ***	−0.025	−0.025	0.055
2019	0.137	0.006	2.883 ***	−0.025	−0.024	0.056
2020	0.041	0.129	1.160	−0.025	−0.023	0.056

Note: *** and ** respectively represent significance levels of 1% and 5%.

**Table 4 ijerph-19-08702-t004:** Estimation results of the panel data model.

Variables	Contribution Rate of GASTP		
	FE	RE	OLS
Per capita GDP	−0.20 **	−0.08	0.11 **
	(−0.09)	(−0.93)	(−0.04)
Agricultural financial support	−0.19 *	−0.06	0.77 *
	(−0.10)	(−0.49)	(−0.42)
Total retail sales of social consumer goods	−0.05	−0.1	0.08 ***
	(−0.05)	(−1.52)	(−0.03)
Total exports	0.00	0.00 ***	0.00 ***
	0.00	(−4.18)	0.00
Planting structure	0.09 ***	0.08 ***	−0.03
	(−0.01)	(−7.19)	(−0.05)
Agricultural mechanization level	−0.02 *	−0.02	0.02 ***
	(−0.01)	(−1.48)	(−0.01)
Constant	3.22 ***	2.23 ***	−0.17
	(−0.71)	(−3.45)	(−0.41)

Note: ***, ** and * represent significance levels of 1%, 5%, and 10%, with t values in parentheses.

## Data Availability

Not applicable.

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
