# Peer review of "Spatio-Temporal Pattern of Green Agricultural Science and Technology Progress: A Case Study in Yangtze River Delta of China"

_ijerph, 2022, doi:10.3390/ijerph19148702_

Round 1

Reviewer 1 Report

In my opinion, the paper can be published in this form. My few recommendations are as follows:

LINES 1-3 I would propose a less long title. For example: Spatio-temporal Pattern of Green Agricultural Science and Technology Progress: a case study in Yangtze River Delta of China

LINE 23 I would insert two more keywords: Food security. Carbon emission

LINES 130-136 it might be useful to justify the choice of evaluation indexes

The research is very specific and the authors themselves should highlight its strengths and weaknesses, as well as possible applications to other contexts. I think they can add these aspects to the discussion. Best regards

Reviewer 2 Report

1.The introduction does not summarize the latest research progress and gaps well.  It is necessary to reorganize the introduction to better elicit research questions of your study. Such as the question of the combination of carbon emission constraints and the contribution rate of GASTP in the study area of YRD. 

2.Reorganize the logic of literature review please. Such as the third paragraph in introduction.

3.Some numbers and letters in Figure 1 are not very clear.

4.Evaluation indexes need to be identified

5.The article draws three main conclusions. Can the discussion be divided into three parts? The discussion section should focus closely on your empirical research results. Please dig deeper into your research results and enrich the discussion content. 

6. Please polish up the language to make the article read more smoothly.
